# Long-Term Stability of Redox Mediators in Carbonate Solvents

**DOI:** 10.3390/molecules27051737

**Published:** 2022-03-07

**Authors:** Felix M. Weber, Ina Kohlhaas, Egbert Figgemeier

**Affiliations:** 1Helmholtz Institute Münster (HI MS), IEK-12, Forschungszentrum Jülich GmbH, Jägerstr. 17-19, 52066 Aachen, Germany; fe.weber@fz-juelich.de; 2Institute for Power Electronics and Electrical Drives (ISEA), RWTH Aachen University, Jägerstr. 17-19, 52066 Aachen, Germany; ina.kohlhaas@isea.rwth-aachen.de; 3Jülich Aachen Research Alliance, JARA-Energy, 52425 Jülich, Germany

**Keywords:** scanning electrochemical microscopy (SECM), redox mediator, 2,5-di-tert-butyl-1,4-dimethoxybenzene (DBDMB), ferrocene, electrolyte

## Abstract

Scanning electrochemical microscopy (SECM) used in the feedback mode is one of the most powerful versatile analytical tools used in the field of battery research. However, the application of SECM in the field of lithium-ion batteries (LIBs) faces challenges associated with the selection of a suitable redox mediator due to its high reactivity at low potentials at lithium metal or lithiated graphite electrodes. In this regard, the electrochemical/chemical stability of 2,5-di-tert-butyl-1,4-dimethoxybenzene (DBDMB) is evaluated and benchmarked with ferrocene. This investigation is systematically carried out in both linear and cyclic carbonates of the electrolyte recipe. Measurements of the bulk current with a microelectrode prove that while DBDMB decomposes in ethyl methyl carbonate (EMC)-containing electrolyte, bulk current remains stable in cyclic carbonates, ethylene carbonate (EC) and propylene carbonate (PC). Ferrocene was studied as an alternative redox mediator, showing superior electrochemical performance in ethyl methyl carbonate-containing electrolytes in terms of degradation. The resulting robustness of ferrocene with SECM is essential for a quantitative analysis of battery materials over extended periods. SECM approach curves depict practical problems when using the decomposing DBDMB for data acquisition and interpretation. This study sheds light towards the use of SECM as a probing tool enabled by redox mediators.

## 1. Introduction

Scanning electrochemical microscopy (SECM), among other scanning probe techniques, is one of the most powerful tools in the field of battery research [1]. Recently, SECM used in the feedback mode has received significant attention in the field of lithium-ion batteries (LIBs) [2]. SECM is a probing method that uses electrodes with a diameter of nano- or micrometers, usually made from platinum, referred to as the tip and is applied in a liquid medium. These microelectrodes are used in close proximity to the sample and a voltage is applied between the tip and counter electrode. In the so-called feedback mode, a redox mediator is added to the solution and a voltage is applied for oxidation. The redox mediator is oxidized at the tip according to Equation (1) and reduced again at the counter electrode.
R → O + n e^−^(1)

Through the continuous oxidation and reduction of the mediator and its diffusion between the two electrodes, a measurable current is recorded. The diffusion-limited current *i_∞_* can be derived from the Cottrell equation and depends on the number of exchanged electrons *n*, the Faraday constant F, the diffusion coefficient of the mediator in the solution *D*, the concentration of the redox mediator *C^*^* and the radius of the tip surface *r*_0_ [3].
*i**_∞_* = 4*n*F*DC**^*^**r*_0_(2)

For the successful use of the feedback mode, a suitable redox mediator needs to be selected. A redox mediator has to be reducible or oxidizable at the tip, may undergo the reverse reaction at the counter electrode [4] and must be active in the stability window of the used liquid. For analysis of changes in the surface properties of battery materials over several hours, the redox mediator is required to enable measurements in this timespan in the electrolyte used.

In recent years, different redox mediators were studied and received attention due to their ability to stabilize lithium-oxygen batteries [5,6,7]. Several redox mediators, including ferrocene [8,9,10] and 2,5-di-tert-butyl-1,4-dimethoxybenzene (DBDMB) [11,12,13,14,15], were proposed for the analysis of LIB electrodes. One electron oxidation and reduction reaction is displayed in Figure 1. DBDMB was initially introduced by Chen et al. [16] as an overcharging agent for LIBs. It was then applied by Bülter et al. as the redox mediator for SECM experiments in the field of LIBs [12]. DBDMB is anodically stable up to a voltage of 4.2 V vs. Li, and beyond that, it starts to decompose [17,18]. The decomposition pathways are proposed to involve the polymerization of the benzene ring and the cleavage of the alkoxy bond, though the polymerization is sterically hindered [19]. Still, DBDMB appeared to be sufficiently stable in electrolytes for LIBs [12,20]. The resulting radical upon oxidation is somewhat reactive, but reported to form the original molecule again via recombination. This behavior strongly depends on the electrolyte medium employed [21]. Moshurchak et al. observed a current decay in a lithium-ion cell with DBDMB, stating that the decay was only due to diffusion of the redox mediator [18]. Due to the fast response time of the microelectrode in SECM experiments and its approximately hemispherical geometry, no current decay caused by diffusion may be observed after a time of several milliseconds has passed [22]. Despite the advantages, DBDMB may penetrate the SEI and react with the substrate underneath or even destabilize the SEI [12,23].

In contrast, ferrocene appears not to diffuse through the SEI [23]. The applied potential in SECM experiments strongly depends on the solvent used [1]. However, SECM experiments are performed with a potential between 3.5 and 3.6 V vs. Li for most electrolytes used in LIBs [10,18,24]. The use of ferrocene requires an oxygen-free environment because the ferrocenium ion produced is decomposed with molecular oxygen in organic solvents [25]. In battery electrolytes, ferrocene was found to be 91% reversibly oxidized [26]. However, despite the number of investigations made with both DBDMB and ferrocene, their stability in SECM experiments in the presence of various electrolytes is not yet known in detail, although SECM studies in the timeframe of several hours may be necessary to observe specific processes, e.g., in LIBs [11]. This study presents the evaluation of DBDMB and ferrocene as potential redox mediators to enable SECM used in the feedback mode in different electrolyte solvents, enlisting linear and cyclic carbonates.

## 2. Materials and Methods

The stability of the redox mediators, 2,5-di-tert-butyl-1,4-dimethoxybenzene (DBDMB) and ferrocene, in electrolyte systems of 1 M LiPF_6_ in EC:EMC and EC:PC was analyzed by recording the bulk current of a polarized microelectrode. Furthermore, SECM approach curves towards glass were recorded in the same electrolytes. Paraffin oil (Vaseline Oil, pure, pharma grade, PanReac AppliChem, Darmstadt, Germany) as well as paraffin oil with dissolved ferrocene were used to cover the electrolyte in order to inhibit the evaporation of the electrolyte during the experiments. This allows the measurement of SECM approach curves over several hours [28].

For the experiment in which ferrocene was added to the covering paraffin oil, UV–vis spectroscopy was used to evaluate the required concentration of ferrocene in paraffin oil to inhibit the liquid–liquid extraction of ferrocene into the oil. An amount of 5 mM ferrocene (99%, Alfa Aesar) was added to 20 mL of the 1 M LiPF_6_/EC:EMC electrolyte (LP50, BASF) and covered by 20 mL of paraffin oil.

The electrolyte 1 M LiPF_6_/EC:EMC (1:1 *w*/*w*) was obtained from BASF (LP50) and 1 M LiPF_6_/ EC:PC (1:1 *w*/*w*) was prepared in house in an argon-filled glovebox (H_2_O and O_2_ < 0.5 ppm). EC, PC and LiPF_6_ were obtained from Merck (Selectilyte 99.9%, Darmstadt, Germany), Alfa Aesar (99%, Kandel, Germany) and Aldrich (battery grade ≥99.99%, Darmstadt, Germany), respectively. All chemicals were used as received.

An amount of 5 mM DBDMB (>98%, Chemos) or 5 mM ferrocene (99%, Alfa Aesar), respectively, was added as the redox mediator to the electrolyte. For each experiment, approximately 2 mL of the electrolyte was used.

A platinum coil counter electrode and a 25 µm platinum microelectrode were applied for all experiments (Sensolytics, Bochum, Germany). The microelectrode was cleaned between all measurements by polishing on alumina sandpaper with different grit sizes (5, 1, and 0.3 µm) and then polishing with silicon oxide (0.03 µm) on a cloth. A bipotentiostat with a low-current amplifier from Metrohm Autolab (PGSTAT30 with ECD module) Utrecht, Netherlands was applied. All data were recorded using software from Sensolytics, Bochum, Germany with an enabled Sallen–Key filter with a RC time constant of 5 s.

The bulk current was recorded every 10 s at a constant voltage of 4.1 V vs. Li for DBDMB and 3.45 V vs. Li for ferrocene for 11 h. Cyclic voltammetry (CV) was recorded with a scan rate of 20 mV/s and steps of 2 mV. Owing to the high sensitivity of lithium (Li) towards oxygen and moisture, the experiments were performed in an argon-filled glove box with H_2_O and O_2_ levels < 0.1 ppm. The potentiostat was placed outside the glovebox using gas-tight feed throughs.

SECM approach curves with 5 mM ferrocene or 5 mM DBDMB as the redox mediator in LP50 were recorded towards a glass surface every 65 min (speed 5 µm/s, resolution 0.5 µm, and waiting time 20 ms). The approach curves were automatically stopped after the current changed by 50% for ferrocene-based experiments and, due to the lower current, by 40% for DBDMB. To prevent the evaporation of the electrolyte, the DBDMB-containing solution was covered with paraffin oil. The ferrocene-containing electrolyte was covered with a solution of 9.5 mM ferrocene in paraffin oil.

All experiments were carried out with a lithium reference electrode (lithium foil, 300 µm, Albemarle).

Cyclic voltammetry experiments of ferrocene in an actual battery setup were performed in CR2032 coin cells with a NMC622 cathode from an industry partner and lithium chips (99.95%, LTS Research, Ø = 15.6 mm). Scanning was performed with a scan rate of 66.6 µV/s between 3.0 and 4.2 V vs. Li with 90 µL of the LP50 electrolyte with 100 mM ferrocene.

## 3. Results

This section presents the bulk current measurements with either DBDMB or ferrocene used as the redox mediator. In an attempt to standardize the different measurements, the recorded currents are normalized to the respective initial values.

Figure 2 shows the evolution of the bulk current with time for EC/EMC (1:1) containing 5 mM DBDMB. As can clearly be seen, the bulk current decreased by 80% within eleven hours. Due to the movements of the tip, SECM measurements need to be carried out in an open system. To avoid the evaporation of the electrolyte (thus inducing changes in bulk properties such as conductivity and viscosity) and related measurement errors, paraffin oil was added on top of the electrolyte for the next experiment. After the measurement, the level of the electrolyte was as high as before the measurement, indicating that the use of paraffin oil successfully suppressed the evaporation of the electrolyte (mainly EMC). It is noteworthy to mention that the addition of paraffin oil on top of the electrolyte did not result in a more stable current. Therefore, the decreasing current is not caused by the evaporation of the electrolyte. The current of the measurement with paraffin oil drops faster than the current without the shield. The evaporation of the more volatile solvent, EMC, leads to a lower electrolyte volume, leading to a higher concentration of the redox mediator. The increased concentration results in a higher current.

When replacing the linear carbonate, EMC, with the cyclic carbonate, PC, the decrease in current can be reduced (see Figure 3). The current in the electrolyte utilizing PC as co-solvent only drops by ~30%, compared to 90% with EMC as co-solvent.

It is hypothesized that the decrease in bulk current is linked to decomposition reactions of the DBDMB, including a chemical reaction after oxidation at the tip. In each measurement, a total amount of charge of 5.5 × 10^−5^–8.3 × 10^−5^ C was transferred. Since DBDMB releases one electron upon oxidation, only a small amount of molecules in the order of 10^−10^ mol is oxidized during the measurement. This is a very small amount compared to the total of approximately 10^−5^ mol DBDMB molecules present in approximately 2 mL of the 5 mM DBDMB-containing electrolyte. A possible irreversible oxidation of DBDMB therefore cannot explain the decrease in the current in the EC:EMC-based electrolytes. Since storage of the DBDMB-containing electrolyte over many days did not have an impact on the current during measurements, a chemical reaction between DBDMB and the electrolyte is unlikely and therefore a coupled electrochemical–chemical reaction might be responsible. This coupled reaction may also be indicated by the hysteresis of the DBDMB CV that is visible at voltages above 4.0 V (see Figure 4 magnification) [3]. Although believed to be difficult because of steric hindrance, electropolymerization of DBDMB might occur [19] with linear carbonates. The reaction products can accumulate, resulting in a clogged microelectrode. The steric hindrance that stabilizes the DBDMB in its oxidized state [20] may be more efficient with the cyclic and therefore less flexible PC molecule. As a result, less DBDMB will react with PC compared to EMC, resulting in a more constant current.

As such, DBDMB could be considered as a suitable redox mediator for electrolytes made from cyclic carbonates. State-of-the-art electrolytes for LIBs mostly use a mixture of cyclic (predominantly EC) and linear (EMC) carbonates and thus cannot be analyzed with DBDMB for long timespans.

As an alternative redox mediator, the stability of ferrocene was analyzed in the EC:EMC-based electrolyte. Ferrocene is known to be a relatively stable metal complex [29]. It is active at lower voltages (3.4 V vs. Li, see Figure 4) than DBDMB (4.1 V vs. Li, see Figure 4). Therefore, the potential lies within the electrochemical window of most of the alkyl carbonate electrolytes. Both redox mediators show a plateau at higher potentials than their oxidation potentials, indicating the presence of a diffusion-controlled bulk current. Figure 5 shows the results for the amperometry measurements of ferrocene in EC:EMC. The use of ferrocene leads to a significantly more constant current compared to DMDMB, only dropping by approximately 45% in eleven hours. The paraffin oil shield turned yellow during the measurement, indicating a partial solution of ferrocene in the oil. This reduces the concentration of the redox mediator in the electrolyte, resulting in a lower current. The liquid–liquid extraction of ferrocene from the electrolyte into the covering paraffin oil was prevented by adding ferrocene to the oil. The required concentration was calculated using UV–vis spectroscopy data. With the addition of ferrocene to the paraffin oil, the current was further stabilized, only decreasing by 25% within eleven hours. In a two-electrode setup without the lithium reference electrode, the current decreased even further. The current only dropped by approximately 5% within eleven hours. From this, it can be concluded that ferrocene is stable within detectable limits even in linear carbonate-based electrolytes. With a similar oxidation voltage to DBDMB, it is a very suitable redox mediator for SECM experiments on LIBs and corresponding battery materials.

The difference in the stability of DBDMB in the EC-based electrolyte containing EMC and PC as co-solvents may be attributed to the difference in nature of the two solvents (EMC and PC). It appears likely that the linear carbonate EMC possesses different reaction kinetics upon the reaction with the DBDMB radical than the cyclic PC. Although the use of PC stabilizes the experimental SECM setup (see Figure 3), the use of PC is not possible for anodes in LIBs. Since the use of PC with graphitic anodes results in exfoliation of graphite sheets due to solvent co-intercalation [30], it is not suitable for SECM experiments on graphitic anode materials. Thus, the use of PC as co-solvent will not be further discussed in this article; we will only discuss the electrolyte made from EC:EMC.

To evaluate the impact of the changing current and decomposition on SECM data for surface analysis, repeated approach curves with ferrocene and DBDMB as redox mediators were recorded.

These SECM approach curves were recorded using glass instead of metallic lithium or other battery materials, although the setup is aimed for measurements in the field of battery research.

Glass was used to ensure an inert, flat and insulating sample. This is required for data evaluation as a change in the sample over time would also lead to changes in the approach curves. With a glass substrate, all changes can be attributed to the effects of the redox mediator or the liquid the measurement took place in. The results of this article can be used to enable long-term measurement in the electrolyte on metallic lithium, anodes or cathodes from LIBs. Similar results using this method were shown recently [28].

The resulting approach curves towards the glass surface with DBDMB and ferrocene as redox mediators are depicted in Figure 6. The position of the tip (x-axis) was taken directly from the position of the stepper motors of the SECM device. The approach curves that were recorded with DBDMB are drawn in Figure 6a. All approach curves feature a decreasing current at lower tip positions on the left side of the figure. This negative feedback curve indicates an insulating surface, as expected with a glass sample. The curves recorded with DBDMB at different times differ and are not highly reproducible. The bulk current decreases over time from approximately 4 nA to approximately 2 nA within 5 h. The approach curves are also shifted to the left—closer to the sample. The movement of the approach curves towards the sample can occur due to local compression of the substrate in the previous approach curves or due to movements of the tip in the tip holder. With the glass substrate, local compression is unlikely. It is possible that small movements of the tip occur upon contact with the sample. From the approach curve at 4:20 h, the slope of the approach curve decreases when in close proximity to the sample. This indicates that the tip has come into direct contact with the sample. The following approach curve at 5:25 h is shifted towards the sample, indicating that the tip might have slid in its holder.

In approach curve experiments, direct contact between the tip and the sample is undesired. Therefore, the approach is usually stopped when the current has reached a lower relative threshold. With a decreasing bulk current over time, the absolute value that changes in the feedback area decreases, while the percentage remains constant. With this reduced change in current, uncertainties become more evident and contact is more difficult to avoid. Furthermore, measurement uncertainties hinder the exact quantitative evaluation of the approach curves as any difference may either occur due to an observed change in the substrate surface, or due to the decreasing bulk current, namely an irreversible redox process of the mediator. A stable measurement system is essential for data acquisition.

Figure 6b depicts the same measurements with ferrocene used as the redox mediator. The bulk current fluctuated between 6.5 and 6.8 nA, with no clear trend over time. The gradient of the approach curve in close proximity to the sample remained constant. This indicates that the tip and the sample are not in mechanical contact and the approach curves are not shifted towards the sample. With a constant bulk current and avoidance of direct contact between the tip and the sample, any differences in the approach curves on other samples can be attributed to a changing surface of the sample.

To further evaluate the approach curves, all approach curves were fitted to the equation suggested by Cornut and Lefrou for a completely insulating substrate [31]. As such, the radius of the active part of the electrode, the position of the sample surface and the Rg value are calculated. The position of the sample cannot be measured directly, but fitting the data to the analytical expression allows the calculation of the distance between the microelectrode and the sample. The Rg value is defined as the ratio of the insulator thickness and the radius of the active part of the microelectrode [4]. The resulting parameters are depicted in Figure 7. The change of the sample position indicates the same results that were already discussed above. With DBDMB as the redox mediator, it is more difficult to avoid contact between the tip and the sample. Upon contact, there is the risk that the microelectrode slides in its holder. This would change the apparent surface position and result in the same data as a decreasing sample height. The contact may further damage the tip itself as parts of the very thin tip may break off. Therefore, avoiding contact between the microelectrode and the sample is crucial.

Figure 7b depicts the values for the radius from the fitted approach curves. The tip was manufactured to feature a 12.5 µm radius, but the results from the fitting fluctuate at approximately 6 µm in the experiments for both redox mediators. The fitting of the approach curves gives the radius of the active area of the tip, which is assumed to be spherical. Minor damage to the tip or a platinum part that is not exactly spherical may change the radius. With DBDMB as the redox mediator, the fitted radii differ from 4.9 to 6.4 µm, whereas the radii in the ferrocene-based experiments differ between 5.9 and 6.0 µm. The latter result clearly promises more reproducible and reliable results using ferrocene as the redox mediator compared with using DBDMB.

The Rg value (see Figure 7c) gives the ratio of the total tip radius (platinum and insulating parts) and the radius of the platinum tip. The fitted values remain almost constant when considering the approach curves taken after one hour or later for both redox mediators. For ferrocene, the value at t = 0 h does not differ significantly. The Rg value for the approach curve at t = 0 h with DBDMB as the redox mediator is approximately double that in later experiments.

According to the equation from Cornut and Lefrou [31], the approach curve depends on the Rg value. A relatively low Rg value results in an approach curve that maintains the bulk current until close proximity of the sample. The current then immediately drops to 0 in an ideal case. A very high Rg value results in a substantially different approach curve: the current is already decreasing at a greater distance from the surface compared with a low Rg value. Therefore, the current does not drop sharply to 0, but is slightly decreasing at greater distances, with an increasing gradient when the tip is approaching the surface.

As shown above, the current decreases over time when using DBDMB as the redox mediator. This occurs because the tip is approaching the insulating sample and because the bulk current decreases when using DBDMB as the redox mediator. The decreasing bulk current is recorded in the approach curve at great distances from the sample. As a result, the current in the recorded approach curve decreases during the entire recorded distance. Fitting of the curve now results in a Rg value that is too large.

The effect of the decreasing bulk current reduces over time, as the decrease in the bulk current is not linear (see Figure 2). Therefore, this effect cannot be seen in the other recorded approach curves. As discussed, the resulting low bulk currents in the later approach curves lead to other problems when performing the experiment.

The described high stability of ferrocene in electrolytes consisting of cyclic and linear carbonates enables further possible uses for ferrocene in the field of battery research. During cycling of lithium-ion batteries, ferrocene does not decompose—as such, a characteristic current of ferrocene is observable in CV measurements. The cyclic voltammetry of a NMC622 cathode with 100 mM ferrocene and with the baseline electrolyte is depicted in Figure 8. The observable noise between 3.4 and 4.5 V in the baseline electrolyte always occurs near the onset potential in this setup and is attributed to inaccuracies during voltage control.

At higher voltages than the already described onset potential of 3.2 V (see Figure 4) for ferrocene, the measured current increases, indicating an ongoing reaction of ferrocene in the electrolyte, while the current in the baseline electrolyte remains zero. This characteristic peak may be used as an internal reference in lithium-ion batteries. Most batteries built for electrochemical testing consist of a two-electrode setup. With the typical disadvantage of only one measurable full-cell voltage, the actual potential of the electrodes remains unclear due to the lack of a reference electrode. Therefore, the use of ferrocene as an internal reference might be a feasible and attractive solution.

## 4. Conclusions

SECM redox mediators, 2,5-di-tert-butyl-1,4-dimethoxybenzene (DBDMB) and ferrocene, have been studied in different electrolyte systems, i.e., 1 M LiPF_6_ in EC:EMC and EC:PC. Analysis of the current at a microelectrode over time showed that DBDMB decomposes in EMC-containing electrolytes. However, it remains stable in electrolytes made from cyclic carbonates, namely EC and PC. For electrolytes consisting of linear carbonates, ferrocene is an effective redox mediator, stable within the electrochemical window of most alkyl carbonate-based electrolytes over extended periods of time. Preventing electrolyte evaporation and changing the redox mediator to ferrocene reduced the decrease in bulk current over eleven hours from 90% with DBDMB to approximately 25% with ferrocene. The decreasing current with DBDMB leads to unpredictable effects when recording and evaluating SECM approach curves. The initially fast decreasing current interferes with the approach curve and alters its shape. The effect appears less dramatic during longer measurement times, but the low current that is present after several hours leads to inaccuracies in the recorded current and difficulties in avoiding the direct contact between tip and sample. The use of ferrocene prevents these problems and allows highly reproducible SECM measurements on battery materials. It was further shown that ferrocene may serve as an internal reference in typical battery electrode setups in order to gain deeper knowledge about electrode potential shifts.

## Figures and Tables

**Figure 1 molecules-27-01737-f001:**
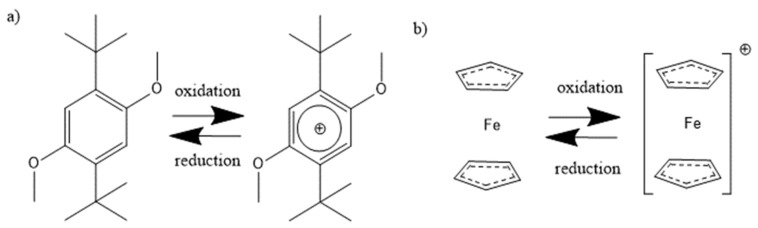
Oxidation and reduction reaction of the redox mediators (**a**) DBDMB [11] and (**b**) ferrocene [27].

**Figure 2 molecules-27-01737-f002:**
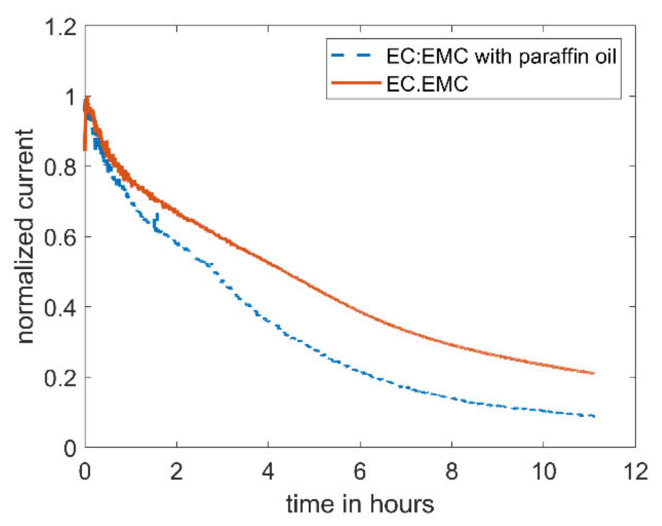
Amperometric current at a microelectrode of 5 mM DBDMB in 1 M LiPF6/EC:EMC with an applied potential of 4.1 V vs. Li. For the lower curve, the electrolyte was covered with paraffin oil during the measurement to avoid evaporation.

**Figure 3 molecules-27-01737-f003:**
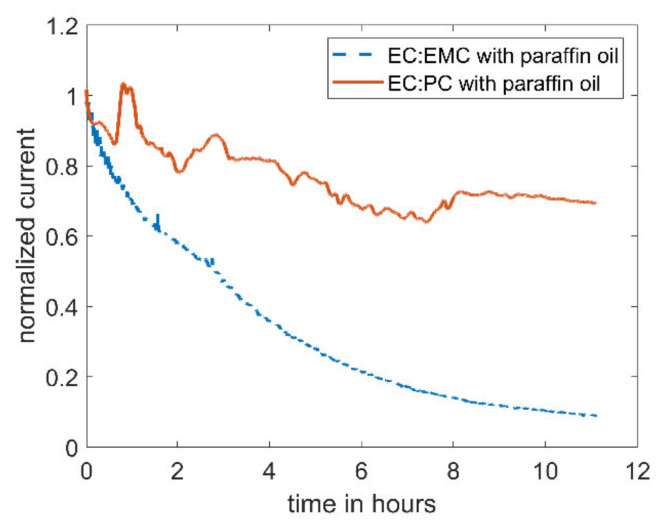
Amperometric current at a microelectrode of 5 mM DBDMB in different electrolytes with an applied potential of 4.1 V vs. Li. In both experiments, the electrolyte was covered with paraffin oil to prevent evaporation. The lower curve corresponds to DBDMB in EC:EMC 1 M LiPF_6_, and the upper curve to DBDMB in EC:PC 1 M LiPF_6_.

**Figure 4 molecules-27-01737-f004:**
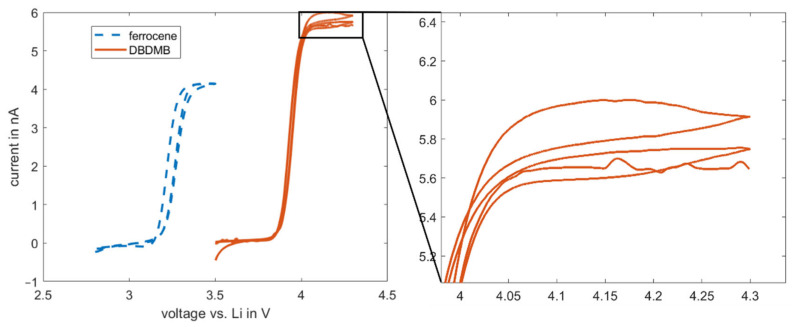
Cyclic voltammetry results for 5 mM DBDMB and 5 mM ferrocene in EC:EMC 1 M LiPF_6_. Measurement was performed with a 25 µm microelectrode. Magnification image is given for the hysteresis at above 4 V for the DBDMB redox mediator. DBDMB and ferrocene are both active in the electrochemical window of most of the electrolytes for LIBs.

**Figure 5 molecules-27-01737-f005:**
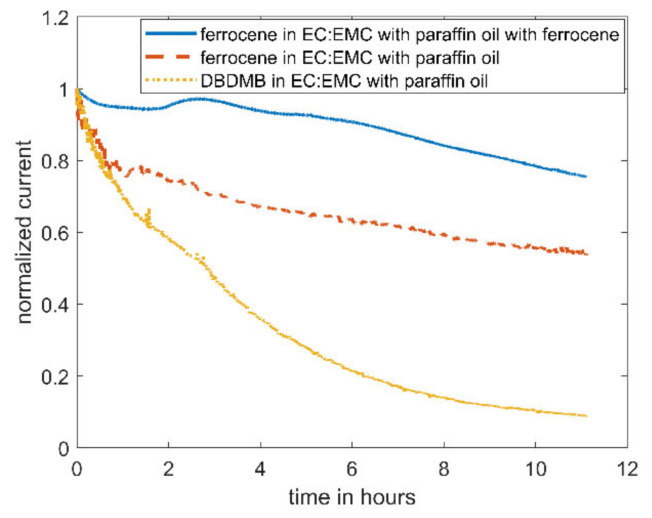
Amperometric current at a microelectrode of 5mM ferrocene in EC:EMC 1 M LiPF_6_ with an applied potential of 4.1 V vs. Li for DBDMB and 3.45 V vs. Li for ferrocene. The middle curve was covered with paraffin oil to prevent evaporation. For the upper curve, 9 mM ferrocene was added to the covering paraffin oil. The lower curve is DBDMB with covering paraffin oil given again for comparison.

**Figure 6 molecules-27-01737-f006:**
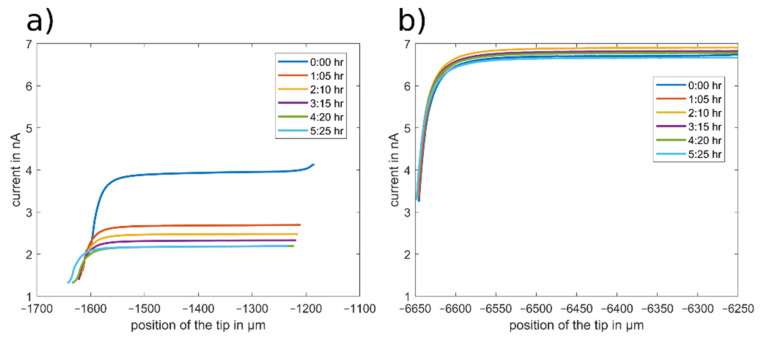
SECM approach curves towards a glass substrate recorded in the LP50 electrolyte at different time steps. The position of the tip (x-axis) is directly taken from the position of the stepper motors. An amount of 5 mM DBDMB (**a**) and 5 mM ferrocene (**b**) were used as redox mediators. A potential of 4.1 V vs. Li was applied when using DBDMB and 3.45 V vs. Li when using ferrocene. A scan rate of 5 µm/s was used.

**Figure 7 molecules-27-01737-f007:**
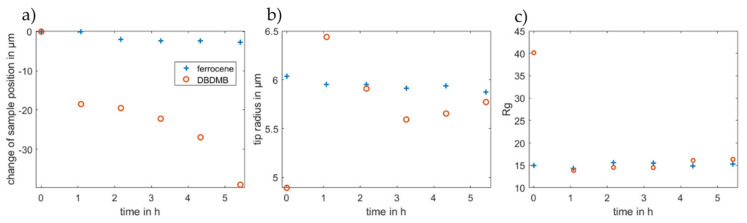
Fitted parameters (**a**) change of sample position, (**b**) tip radius and (**c**) Rg value for the approach curves depicted in Figure 6. Fitting was accomplished using the analytical expression for an insulator according to Cornut and Lefrou [31].

**Figure 8 molecules-27-01737-f008:**
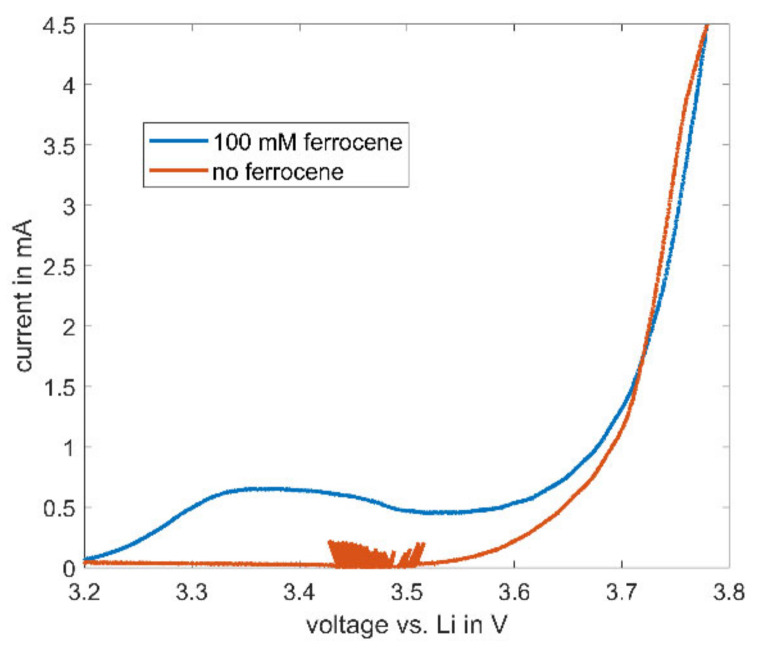
Cyclic voltammetry results for half cells of NMC622 with an electrolyte consisting of EC:EMC (1:1) with 1 M LiPF_6_ and 100 mM ferrocene at increasing voltage. CV measurement was carried out with a scan rate of 66.6 µV/s.

## Data Availability

Not applicable.

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
