# Peer review of "Long-Term Stability of Redox Mediators in Carbonate Solvents"

_molecules, 2022, doi:10.3390/molecules27051737_

Round 1

Reviewer 1 Report

The quality of the work is very good and precise. As you mentioned that the ferrocene is found to be an effective redox mediator, being stable within the electrochemical window of most alkyl carbonate based electrolytes over extended periods of time. 
Prevention of the electrolyte evaporation and the exchange of the redox mediator to ferrocene reduced the decrease in bulk current during eleven hours from 90% with DBDMB to about 25%. Using ferrocene prevents the problems  that occurs during the process of the former works and allows highly reproducible SECM measurements on battery materials.  It deserves to show that it   may serve as an internal reference in typical battery electrode set-ups in order to gain deeper 

Author Response

The authors would like thank the reviewer for the positive evaluation of the manuscript.

Reviewer 2 Report

The authors did a good job.

The electrochemical/chemical stability of 2, 5-di-tert-butyl-1, 4-dimethoxybenzene (DBDMB) and ferrocene is evaluated. Thus they mainly address selection of appropriate redox mediators. The topic is not novel but it is relevant in the field. The authors try to address that much work has been done on those mediators in LIB however their stability in SECM is unknown, so they addressed this in their experiments and showed that the two mediators remained stable in EC and PC.  

UV-vis spectra experiments should be done to verify the stability before and after oxidation. however, the authors should further explain what happen after the oxidation. Do thr alkoxy groups affect the performance of the aromatic redox shuttles hence it is crucial to give details on this.

The references should be add these as well.

A Highly Active Low Voltage Redox Mediator for Enhanced Rechargeability of Lithium–Oxygen Batteries | ACS Central Science

Critical Role of Redox Mediator in Suppressing Charging Instabilities of Lithium–Oxygen Batteries | Journal of the American Chemical Society (acs.org)

Controllable and stable organometallic redox mediators for lithium oxygen batteries - Materials Horizons (RSC Publishing)

The figures are not bright enough, the authors should try and make them thick.

Author Response

The authors did a good job.

The electrochemical/chemical stability of 2, 5-di-tert-butyl-1, 4-dimethoxybenzene (DBDMB) and ferrocene is evaluated. Thus they mainly address selection of appropriate redox mediators. The topic is not novel but it is relevant in the field. The authors try to address that much work has been done on those mediators in LIB however their stability in SECM is unknown, so they addressed this in their experiments and showed that the two mediators remained stable in EC and PC.  

The authors would like to thank the reviewer for the positive evaluation of the submitted manuscript and the helpful suggestions that will be answered in the following:

UV-vis spectra experiments should be done to verify the stability before and after oxidation. however, the authors should further explain what happen after the oxidation. Do thr alkoxy groups affect the performance of the aromatic redox shuttles hence it is crucial to give details on this.

We thank the reviewer for the suggestions on the experimental set-up. The reaction of the redox mediator after its oxidation remains unclear at this point. We already discussed possible reactions in the manuscript from what has been published in other literature (see line 170-187). With our techniques no identification of the ongoing chemical reaction is possible. We acknowledge the reviewer to point out the lacking reaction mechanism and this will certainly be of high interest in our research and will be addressed in the future. Still, the focus of this publication is on the application and (non-)stability of the redox mediators, which we think is of highest importance. In regard to the main message of our contribution, the underlying chemical mechanisms of degradation products are of secondary interest.

The references should be add these as well.

A Highly Active Low Voltage Redox Mediator for Enhanced Rechargeability of Lithium–Oxygen Batteries | ACS Central Science

Critical Role of Redox Mediator in Suppressing Charging Instabilities of Lithium–Oxygen Batteries | Journal of the American Chemical Society (acs.org)

Controllable and stable organometallic redox mediators for lithium oxygen batteries - Materials Horizons (RSC Publishing)

We thank the reviewer for suggesting the additional published literature and have added the referenced into the manuscript.

The figures are not bright enough, the authors should try and make them thick.

We thank the reviewer for pointing this out and have increased the line thickness for better readability.